# Curcumin Reprograms TAMs from a Protumor Phenotype towards an Antitumor Phenotype via Inhibiting MAO-A/STAT6 Pathway

**DOI:** 10.3390/cells11213473

**Published:** 2022-11-02

**Authors:** Mingjing Jiang, Ying Qi, Wei Huang, Ying Lin, Bo Li

**Affiliations:** 1Liaoning Provincial Key Laboratory of Oral Diseases, Experimental Teaching Center, School and Hospital of Stomatology, China Medical University, Shenyang 110001, China; 2Jilin Provincial Key Laboratory of Oral Biomedical Engineering, Department of Oral Anatomy and Physiology, Hospital of Stomatology, Jilin University, Changchun 130021, China

**Keywords:** immunotherapy, repolarization, curcumin

## Abstract

M1 phenotype macrophages have anticancer characteristics, whereas M2 phenotype macrophages promote tumor growth and metastasis. A higher M1/M2 ratio, therefore, has a beneficial effect on the tumor immune microenvironment, thereby inhibiting tumor growth. The natural alkaloid curcumin is found to have anticancer properties. However, the mechanism remains unclear. In this study, a cell co-culture system and M2 macrophage model were used to evaluate the effects of curcumin on tumor-associated macrophage (TAM) phenotypes. Our results demonstrate that curcumin reprogrammed the M2 macrophages by reducing the level of anti-inflammatory cytokines (TGF-β, Arg-1, and IL-10) and an M2 surface marker (CD206) induced by Cal27 cells or IL-4, as well as upregulating proinflammatory cytokines (TNF-α, iNOS, and IL-6) and an M1 surface marker (CD86). The in vitro assays suggested that curcumin treatment suppressed the migration and invasion of the Cal27 cells induced by the M2-like macrophages. Mechanistically, the repolarization of TAMs may be attributed to the inhibition of monoamine oxidase A (MAO-A)/STAT6 signaling after curcumin treatment. Collectively, our results show that the anticancer effects of curcumin could be explained by reprogramming TAMs from a protumor phenotype towards an antitumor phenotype.

## 1. Introduction

A growing number of people worldwide are suffering from oral squamous cell carcinoma (OSCC). Despite the fact that multiple treatments are available for OSCC patients, the five-year survival rate remains low. Among OSCC patients, metastases and recurrences are the main causes of death. Understanding the underlying mechanisms of OSCC development and occurrence is vital to finding new treatment options.

Tumor-associated macrophages (TAMs) are mainly classified into the M1 and M2 phenotypes [1]. Ample evidence suggests that TAMs in the tumor microenvironment (TME) are predominantly the M2 phenotype and are conducive to tumor invasion, metastasis, and angiogenesis by interacting with cancer cells [2]. However, TAMs are not fixed in polarization. TAMs are capable of changing their phenotypes in solid tumors, which allows them to be therapeutically useful [3,4]. Consequently, repolarizing TAMs from having immunosuppressive and tumor-promoting characteristics to having immunostimulatory and tumor-killing characteristics has gained momentum in immunotherapy [5]. Several reagents, such as monoamine oxidase A (MAO-A) inhibitors, proteasome inhibitors, and regorafenib, have been shown to improve cancer immunotherapy by reprogramming TAMs [6,7,8]. A major objective of current cancer immunotherapy is to identify new molecules that regulate TAM polarization and to develop new combinations aimed at reprogramming TAMs [9].

It is known that curcumin, one of the active ingredients in *Curcuma longa*, can be used for the treatment of chronic diseases, including cancer. There has been a vast amount of evidence demonstrating curcumin’s potency as a tumor suppressor in recent decades [10]. According to research, curcumin inhibited cell proliferation and induced ferroptosis in breast cancer by increasing SLC1A5 expression [11]. Curcumin therapy exerted multiple antitumor activities by modulating signaling pathways associated with cancer development, including the Hedgehog/Gli1, PI3K/AKT, NF-kB, and Wnt/β-catenin pathways [12,13,14]. In addition, clinical studies showed that curcumin was effective in inhibiting tumor growth [15]. Moreover, curcumin exhibited a good safety profile, without significant side effects, from the oral administration of high doses of curcumin [16,17]. However, it remains poorly understood whether the antitumor effect of curcumin is associated with TAMs.

In this study, we found that curcumin was effective in inducing M1-like macrophage properties in the M2 macrophages. These repolarized M2 macrophages exhibited a higher level of IL-6, TNF-α, CD86, and iNOS and enhanced the inhibition of the migration and invasion behavior of OSCC cells in vitro. Mechanistically, curcumin treatment downregulated the expression of MAO-A and suppressed the STAT6 pathway. In summary, we demonstrate that curcumin can promote the polarization of TAMs towards the M1 phenotype via the inactivation of the MAO-A/STAT6 pathway. These findings identify curcumin as a key modulator of TAMs, and reprogramming TAMs with curcumin holds promise for improving cancer immunotherapy.

## 2. Materials and Methods

### 2.1. Cell Culture and Polarization of Macrophages

Human OSCC line Cal27 and murine macrophage-like cell line Raw264.7 were bought from the China Center for Type Culture Collection (CCTCC). These cells were grown in Dulbecco’s modified Eagle’s medium (DMEM) supplemented with 10% fetal bovine serum (FBS), 100 U/mL of penicillin, and 100 μg/mL of streptomycin at 37 °C in a humidified atmosphere containing 5% CO_2_. The supernatant of the culture was collected and centrifuged at 3000× *g* for 5 min to remove the cells. Then, the supernatant was filtered through a 0.22 μm filter to eliminate cell debris. IL-4 (20 ng/mL, PeproTech, East Windsor, NJ, USA) was employed to induce Raw264.7 differentiation into M2 subset macrophages.

### 2.2. Cell Proliferation Assay

Raw264.7 cells were plated in 96-well plates at 5 × 10^3^ cells per well at the logarithmic phase and stimulated by IL-4 (20 ng/mL) for 24 h, or by a conditional medium of Cal27 cells (CM-Cal27) for 48 h. Then, the cells were exposed to curcumin at various concentrations (2.5, 5, 10, 20, 40, and 80 µM). After 24 h, 10 µL of Cell Counting Kit-8 (CCK-8) reagent (APEXBIO, Houston, TX, USA) was added to each well, and the plates were then incubated at 37 °C in the dark for 1 h. The absorbance at 450 nm was determined using a microplate reader (Tecan, Mechelen, Belgium).

### 2.3. Co-Culture Experiment

The bottom layer of a 6-well growth plate was seeded with Raw264.7 cells (2 × 10^5^ per well). Cal27 cells (5 × 10^5^ per well) were planted in a 0.4 µm insert layer of each well (Labselect). The following day, Cal27 cells in the insert layer and Raw264.7 cells in the bottom layer were combined for co-culture.

### 2.4. Transwell Assay

For the Transwell migration experiment, Cal27 cells (2 × 10^4^ per well) in 200 µL serum-free media were introduced to the upper chambers of a 24-well Transwell plate (pore size, 8 mm; Corning, NY, USA). Raw264.7 cells were added to the lower chambers in 600 mL of medium with 10% FBS and co-cultured with the upper compartment cells for 24 h. The cells that had migrated through the membrane were fixed in 4% paraformaldehyde for 15 min and stained with 0.2% crystal violet dye for 20 min. Then, the upper chambers were washed with distilled water and photographed under the microscope. The cells that had traversed the membrane were counted in five different fields (200×). For the Transwell invasion assay, Matrigel (Corning, NY, USA) was utilized to precoat the upper chamber of the Transwell plates for 2 h, and Raw264.7 cells and Cal27 cells were co-cultured for 48 h.

### 2.5. Wound-Healing Assay

Cal27 cells were seeded in 6-well plates. After the cells had a confluence of around 90%, they were wounded by scraping with a 200 µL pipette tip twice per well and washed 3 times with PBS. Cal27 cells were then treated with different supernatants from Raw264.7 cells. Then, at 0 h and 24 h, eight sites of each well with 100× magnification were photographed through microscopy. ImageJ software was used to analyze the wound’s area. The healing rate was calculated as (%) = (initial average scratch area—average scratch area at 24 h)/initial average scratch area × 100%.

### 2.6. Quantitative Real-Time PCR

Trizol (Takara Bio Inc., Dalian, China) was used to extract total cellular RNA, and the PrimeScriptTM RT Reagent Kit (Takara Bio Inc., Dalian, China) was used to reverse transcribe the RNA into cDNA. Then, the TB Green Premix Ex Taq II kit (Takara Bio Inc., Dalian, China) was used to carry out quantitative real-time PCR (qRT-PCR) with a 7500 Real-time PCR system. Table 1 contains a list of the primers utilized in this study.

### 2.7. Western Blot Analysis

Raw264.7 cells were cultured with complete DMEM containing IL-4 (20 ng/mL) for 24 h to differentiate into the M2 macrophages, which were further treated with curcumin (20 µM) for 24 h. Then, the cells were harvested and lysed with RIPA lysis buffer (Beyotime, Jiangsu, China, P0013B) containing PMSF (Phenylmethanesulfonyl fluoride, Beyotime, ST505) and phosphatase inhibitor cocktail A (Beyotime, P1081) to collect total protein. The 50× store concentration of phosphatase inhibitor cocktail A includes 250 mM sodium fluoride, 50 mM sodium pyrophosphate, 50 mM β-glycerophosphate, and 50 mM sodium orthovanadate. The collected total protein was kept on ice for 1 h. Then, the cell lysis buffer was centrifuged at 12,000× *g* for 15 min at 4 °C. The extracted supernatant was used for protein quantification using a BCA Protein Assay Kit (Beyotime, P0012S) first. Equal amounts of the protein were loaded and separated by 10% sodium dodecyl sulfate-polyacrylamide gel electrophoresis (SDS-PAGE) and then transferred to a PVDF Membrane (Millipore, Temecula, CA, USA). The membranes were blocked with nonfat milk for 1 h and incubated with anti-β-actin (Abcam, Cambridge, UK; ab115777, 1:2000 dilution), E-Cadherin (Abcam; ab40772, 1:10,000 dilution), anti-N-Cadherin (Abcam; ab76011, 1:5000 dilution), anti-Vimentin (Abcam; ab92547, 1:2000 dilution), anti-STAT6 (Abcam; ab263947, 1:1000), anti-P-STAT6 (Abcam; ab32520, 1:2000), and anti-MAO-A (Abcam; aab126751, 1:10,000 dilution) overnight at 4 °C. Then, the membranes were washed and incubated with the secondary antibody Dylight 800 Goat Anti-Rabbit IgG (Abbkine; A23920, 1:2000). Finally, protein bands were visualized using an Odyssey CLX (LI-COR, Lincoln, NE, USA). Computer images of bands from Western blot were analyzed using ImageJ software, and the ratio between the pixel’s gray value of the target band and reference band was applied to the statistical analysis.

### 2.8. Enzyme-Linked Immunosorbent Assay (ELISA)

According to the manufacturer’s instructions, ELISA was performed. The Mouse IL-6 ELISA KIT (CSB-E04639m) and Mouse TNF-α ELISA KIT (CSB-E04741m) were purchased from CUSABIO. The Mouse IL-10 ELISA kit (ELM-IL10-1) was purchased from RayBiotech Inc. (Peachtree Corners, GA, USA).

### 2.9. Flow Cytometry

After stimulation with IL-4 (20 ng/mL) or co-culture with Cal27 cells, Raw264.7 cells were incubated in the medium with curcumin (20 µM) for 24 h. Then, a single-cell suspension of the Raw264.7 cells was prepared and stained with the following fluorochrome-conjugated antibodies in PBS containing 1% BSA on ice for 20 min at 4 °C according to the dilution ratio recommended by the manufacturers: anti-mouse APC-CD86, anti-mouse FITC-CD20, and anti-mouse PE-Sirp-α. Fluorescence-activated cell sorting (FACS) was used to measure the stained cells. The measurement wavelength of CD206 and Sirp-α is blue laser (488 nm), and the measurement wavelength of CD86 is red laser (633 nm). Data analysis was performed using FlowJo software.

### 2.10. Reactive Oxygen Species (ROS) Measurement

After inducing differentiation using IL-4 (20 ng/mL) for 24 h, curcumin (20 µM) was added to stimulate Raw264.7 cells for another 24 h. Then, Raw264.7 cells (1 × 10^6^ cells) were collected and resuspended in 1 mL DMEM that contained 1 µM DCFH-DA (Beyotime, S0033M). After 20 min of incubation at 37 °C, the cells were immediately washed with DMEM three times and subjected to flow cytometry analysis. Intracellular ROS can oxidize nonfluorescent DCFH to generate fluorescent DCF. The fluorescence intensity of DCF fluorescence probes was measured using a flow cytometer, which indirectly reflected the level of ROS.

### 2.11. Statistical Analysis

The GraphPad Prism 7.0 software was applied to analyze the statistical data, and differences between the groups were compared using Student’s *t*-tests or analysis of variance (ANOVA). The results are expressed as the mean ± SD, and a *p* value less than 0.05 indicates a statistically significant result.

## 3. Results

### 3.1. Curcumin Promoted the Expression of Antitumor Cytokines and Suppressed the Expression of Protumor Cytokines in CM-Cal27-Induced Raw264.7 Cells

While curcumin can suppress tumor development and progress, it is still unclear whether curcumin inhibits the malignant behaviors of OSCC through modulating macrophage-mediated immune responses. To investigate the impact of curcumin on TAMs in the TME, the Raw264.7 cells were first exposed to the conditional medium of the Cal27 cells (CM-Cal27) to induce TAM differentiation [18,19]. As shown in Figure 1A,B, CM-Cal27 simultaneously upregulated the relative mRNA expression of anti-inflammatory and proinflammatory cytokines in the Raw264.7 cells, suggesting that the Raw264.7 cells may undergo differentiation towards TAMs. The results of the CCK-8 assay showed that the safe concentrations of curcumin were 5, 10, and 20 µM, which were used in the below experiments (Figure 1C). Then, qRT-PCR and ELISA were employed to compare the levels of cytokines in the CM-Cal27-induced Raw264.7 cells after curcumin treatment. Our results show that curcumin promoted the expression of antitumor cytokines (TNF-α and IL-6) and decreased the level of protumor cytokines (IL-10) in the CM-Cal27-incubated Raw264.7 cells at a concentration of 20 µM (Figure 1D,E). These results indicate that 20 µM curcumin may promote the transformation of the CM-Cal27-induced Raw264.7 cells towards an antitumor phenotype. However, the FACS results showed that curcumin affects neither the expression of the M1 macrophage surface marker CD86 nor the M2 macrophage surface marker CD206 (Figure 1F,G).

### 3.2. Curcumin Facilitated TAMs Polarization into an M1-like Population in the Co-Culture System

To observe the effect of curcumin on the M1 and M2 phenotypes, we used an in vitro co-culture system to acquire TAMs according to previous reports [20,21]. The Raw264.7 cells were seeded in the lower chambers and co-cultured with the Cal27 cells in the upper chambers first. Then, the Cal27 cells were removed, and the Raw264.7 cells in the lower chambers were treated with curcumin (20 µM). After 24 h, the Raw264.7 cells were collected for further analysis. We found that curcumin treatment decreased IL-10, Arg-1, and CD206 and increased IL-6, TNF-α, and iNOS expression at the mRNA level (Figure 2A). The expression of the M2 macrophage surface marker CD206 and the M1 macrophage surface marker CD86 was measured through flow cytometry analysis. The in vitro co-culture system could motivate Raw264.7 to express CD86 and CD206 first. However, after treatment with curcumin, CD86 was increased, and CD206 was decreased considerably (Figure 2B–E). These results suggest that curcumin could promote the transformation of TAMs towards an M1 phenotype and suppress M2 polarization. To further identify whether curcumin had the same functions in the TME accompanied by crosstalk between cancer cells and TAMs, we co-cultured the Raw264.7 cells with the Cal27 cells for 24 h in the same way. Then, curcumin was added into the co-culture system without removing the Cal27 cells in the upper chambers. After 24 h, flow cytometry analysis was used to measure the percentage of CD86- and CD206-positive populations in the Raw264.7 cells. In agreement with previous studies (Figure 2B–E, and Appendix A), curcumin significantly promoted the expression of the membrane protein CD86 and reduced the expression of CD206 (Figure 2F–I, and Appendix A). This evidence demonstrates that curcumin facilitated TAMs’ polarization into an M1-like population in a co-culture cell system.

### 3.3. Curcumin Reprogrammed M2 Macrophages into an M1-like Population in IL-4-Induced Model

IL-4 can induce macrophage polarization to an immunosuppressive M2 phenotype [22]. We adopted this type of M2-phenotype macrophage in our experiments to further demonstrate whether curcumin can facilitate the shift of TAMs from an M2 phenotype to an M1 phenotype. We first treated the Raw264.7 cells with IL-4 (20 ng/mL) to induce the M2 macrophages, which were subsequently stimulated by curcumin (20 µM). Representative micrographs of morphologic changes of Raw264.7 cells were shown in Appendix A. We found that curcumin resulted in the upregulation of the mRNA expression of the M1 macrophage markers TNF-α, IL-6, IL-12, CD86, and iNOS, and contributed to a decline in the mRNA expression of the M2 macrophage markers IL-10, CD206, TGF-β, and Arg-1 (Figure 3A,B). Additionally, curcumin reduced the protein expression of Arg-1 (Figure 3C,D), increased the protein level of iNOS (Figure 3C,E), and induced the secretion of IL-6 and TNF-α in the M2 macrophages (Figure 3F). Even more to the point, curcumin treatment resulted in the highly downregulated expression of CD206 and increased the percentage of the CD86-positive population (Figure 3G−J, and Appendix A) in the M2 macrophages induced by IL-4. These findings are consistent with the results in Figure 2 and show that curcumin helped repolarize the M2 phenotypes into an M1-like population, which is supported by the elevated levels of proinflammatory and antitumor cytokines, as well as the reduced levels of anti-inflammatory and protumor cytokines caused by IL-4.

### 3.4. Curcumin Reprogrammed M2 Macrophages from a Protumor Phenotype towards an Antitumor Phenotype

Epithelial-mesenchymal transition (EMT) is an important underlying mechanism of primary tumorigenesis and metastasis [23]. During EMT, epithelial cells will convert from epithelial cells highly expressing epithelial markers (E-cadherin) to mesenchymal cells acquiring mesenchymal markers (N-cadherin and Vimentin). This transition can impair the adhesion of epithelial cells and enhance motility [24]. It has been reported that M2 macrophage-derived cytokines can accelerate the invasion and metastasis of OSCC cells [25]. The previous results demonstrate that curcumin reprogrammed TAMs towards an M1 phenotype and regulated the expression of cytokines. Whether curcumin could block the tumor-promoting effects of TAMs by altering their phenotypes needs to be explored. Firstly, we measured the modulatory effects of curcumin on the invasive and migratory ability of tumor cells. The results of the Transwell assay suggest that the M2 macrophages promoted the cell migration and invasion properties of the Cal27 cells, while curcumin attenuated these abilities of the Cal27 cells through repolarizing the M2 macrophages (Figure 4A–D). The 24-h wound-healing assay was also employed to further measure the migration ability of the Cal27 cells. The healing rate of the Cal27 cells induced by the supernatant from the curcumin-pretreated M2 macrophages was decreased compared to the control group (Figure 4E,F). Furthermore, the epithelial markers (E-cadherin) and mesenchymal markers (N-cadherin and Vimentin) in the Cal27 cells were detected by qRT-PCR and Western blot. The results of Western blot show that curcumin increased the protein level of E-cadherin (Figure 4G,H). The above results reveal that curcumin inhibited the Cal27 cells’ migration and invasion capacity induced by the M2 macrophages through promoting M2 macrophage transformation towards an M1 phenotype.

Macrophages can directly phagocytose tumor cells to exert antitumor effects. M1 macrophages have a strong phagocytic capacity, while TAMs mainly exhibit an M2 phenotype with impaired phagocytosis during tumor progression [8,9]. We next evaluated the effects of curcumin on the phagocytosis of TAMs via measuring the expression of several classical phagocytosis checkpoints: THBS1, Sirp-α, and Siglec-G [26,27,28]. We found that curcumin inhibited the gene level of Sirp-α, THBS1, and Siglec-G (Figure 4K) and decreased the cell surface level of SIRP-α in the M2 macrophages (Figure 4I,J), which may be associated with the enhancement of the phagocytic capacity of macrophages. Taken together, these findings demonstrate that curcumin reprogrammed TAMs towards an antitumor phenotype.

### 3.5. Curcumin Reprogrammed M2 Macrophages towards M1-like Macrophages by Suppressing MAO-A/STAT6 Signaling

Besides its anticancer application, curcumin has other biomedical applications such as anti-inflammatory, antioxidant, neuroprotective, and antidepressant effects. The antidepressant activity of curcumin was found to be associated with monoamine oxidase (MAO) suppression [29]. Curcumin was also found to inhibit MAO-A [30]. In particular, a recent study evaluated the role of MAO-A in regulating macrophage polarization. The results demonstrated that MAO-A facilitated TAMs’ immunosuppressive polarization via upregulating ROS and activating the STAT6 pathway [6]. We postulated that curcumin may promote M1 macrophage polarization by suppressing MAO-A and thereby inactivating STAT6 signaling.

To test this hypothesis, we measured the MAO-A level in the M2 macrophages treated with or without curcumin. Our results show that curcumin treatment gave rise to the downregulation of MAO-A (Figure 5A–C). In agreement with the previous study that found that high-level MAO-A upregulated ROS to promote macrophage immunosuppressive polarization, curcumin was able to simultaneously downregulate MAO-A and ROS levels in the M2 macrophages (Figure 5F). The activation of the JAK-STAT6 signaling pathway was found to be of great importance to the immunosuppressive polarization of macrophages [31,32]. After IL-4 stimulation, STAT6 undergoes phosphorylation and nuclear migration to bind to genes associated with the immunosuppressive function of macrophages and induce their transcription [33]. ROS has also been verified to enhance the phosphorylation of JAK/STAT6 in a great diversity of cells. Since the downregulation of MAO-A and ROS was observed in the M2 macrophages after curcumin treatment, we postulated that curcumin may promote M1 polarization via MAO-A/ROS downregulation-related STAT6 signaling suppression. Indeed, curcumin reduced STAT6 phosphorylation in the M2 macrophages (Figure 5B,E, and Appendix A). These data suggest that curcumin may be capable of re-educating macrophages from a protumor phenotype towards an antitumor phenotype via downregulating the M2 macrophages’ intracellular MAO-A/ROS and thereby suppressing STAT6 signaling.

## 4. Discussion

TAMs, which can be categorized as proinflammatory M1 macrophages and anti-inflammatory M2 macrophages, are crucial in the development and metastasis of cancers [34,35]. TAMs make up over 50% of the immune cells in the TME. The M2 phenotype, which is associated with increased tumor cell migration and invasion [36], vascular distribution [37], and drug resistance [38], is a principal factor of tumor progression. TAMs also weaken antitumor immunity in many ways by, for example, promoting the development of regulatory T cells, which further exacerbates the immunosuppressive tumor microenvironment [39,40]. On the other hand, a tiny population of M1 macrophages mediates proinflammatory activities in the TME and is essential for antitumor immunity [41]. Plenty of preclinical investigations have largely demonstrated that M2 macrophages play a tumor-promoting role in solid tumors. Targeting these protumor characteristics of TAMs may also be a promising therapeutic approach. The main approaches in the endeavor to develop macrophage-targeted cancer therapeutics are macrophage exhaustion, macrophage recruitment blocking, and macrophage polarization towards an antitumor phenotype [42].

It has been shown that curcumin can block EMT in oral cancer cells [43], inhibit tumor cell invasion and metastasis [44], resist proliferation, and promote apoptosis to suppress OSCC progression [45]. However, the precise antitumor mechanism of curcumin remains unclear. The infiltration of TAMs has been reported to lead to malignant behaviors of cancer cells [46,47], tumor growth, and metastasis [48]. Whether curcumin can regulate the tumor immune microenvironment and phenotypes of TAMs, the main immune cell population in the TME, needs to be confirmed. In our study, we hypothesized that curcumin could promote the transition of TAMs from a protumor phenotype to an antitumor phenotype, thus reversing the EMT of cancer cells and suppressing OSCC invasion and migration induced by TAMs. To test this hypothesis, Raw264.7 cells stimulated by CM-Cal27 or IL-4, or co-cultured with the Cal27 cells, were treated with curcumin. In Figure 1, curcumin promoted the expression of antitumor cytokines (TNF-α and IL-6) and decreased the level of protumor cytokine (IL-10) in the CM-Cal27-incubated Raw264.7 cells at a concentration of 20 µM, but curcumin affects neither the expression of CD206 nor CD86. Therefore, the co-culture system and M2 macrophage model were used to further confirm whether curcumin could change the phenotype of TAMs (Appendix A). Our results demonstrate that curcumin reduced the production of anti-inflammatory cytokines and the M2 surface marker CD206 induced by IL-4 or the Cal27 cells, and significantly upregulated proinflammatory cytokines and the M1 surface marker CD86. This evidence supports our hypothesis that curcumin could repolarize TAMs from an M2 phenotype to an M1 phenotype (Figure 2 and Figure 3).

TAMs can upregulate the expression of EMT markers (E-cadherin, N-cadherin, and vimentin) and promote the migration and invasion of cancer cells with the participation of many regulators [49]. The results presented in this study demonstrate that curcumin repolarized TAMs towards an M1 phenotype and inhibited the expression of anti-inflammatory cytokines. Further work evaluated the effect of curcumin on the migration and invasion abilities of OSCC cells induced by macrophages. The Transwell assay verified that the M2 macrophages promoted the migration and invasion of the Cal27 cells, and curcumin could attenuate these abilities of the Cal27 cells through repolarizing the M2 macrophages. The wound-healing assay consistently found that the supernatant from the M2 macrophages treated with curcumin decreased the migration of the Cal27 cells and increased the expression of E-cadherin in the Cal27 cells. These results reveal that curcumin could inhibit the migration and invasion capacity of the Cal27 cells by re-educating the M2 macrophages towards an M1 phenotype. To further explore the effect of curcumin on macrophages, we also tested the expression of phagocytosis “checkpoints”, such as THBS1, Siglec-G, and SIRP-α, which are involved in suppressing macrophage phagocytosis against cancer cells. We found that curcumin decreased the gene expression and cell surface level of SIRP-α. Taken together, curcumin led to an enhancement of the antitumor ability of macrophages via facilitating M1 polarization.

Previous studies showed that curcumin had potent inhibitory activity towards MAO-A in the management of depression [30]. The latest research found that MAO-A played a critical role in promoting TAMs’ immunosuppressive polarization, and MAO-A inhibitors repolarized TAMs to an immunostimulatory phenotype through suppressing STAT6 signaling [6]. In addition, SOCS1 inhibition and STAT6 activation can promote the polarization of M2-subtype macrophages, enhance the invasion and migration ability of OSCC cells, and accelerate tumor growth [50]. In our study, we observed that curcumin treatment decreased the level of MAO-A and its downstream product, intracellular ROS, and thereby inactivated STAT6 signaling. This indicates that curcumin may be able to re-educate M2 macrophages towards an antitumor phenotype via downregulating intracellular MAO-A/ROS levels and suppressing the STAT6 signaling pathway. Overall, the present study supports our efforts to research the therapeutic effects of curcumin and its antitumor mechanism in vivo. This study may offer a novel perspective on the therapeutic potential of curcumin for OSCC and perhaps other malignancies as well.

## Figures and Tables

**Figure 1 cells-11-03473-f001:**
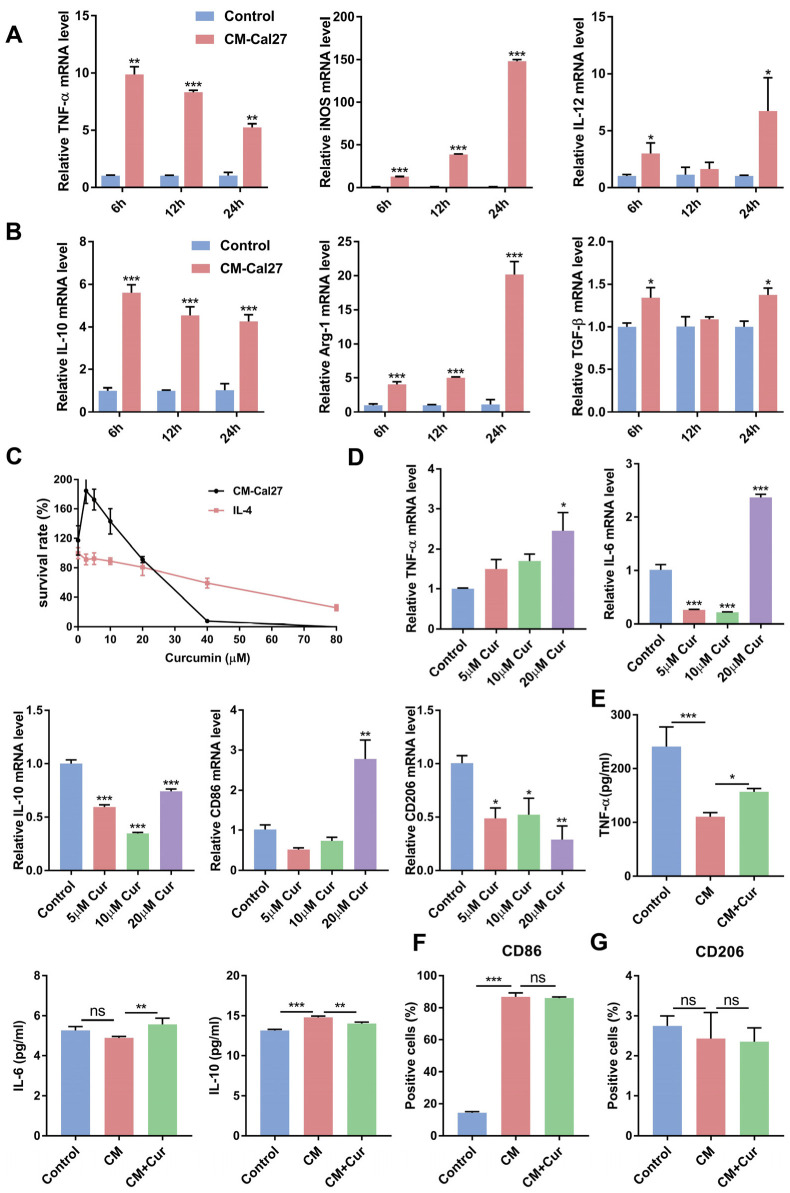
Curcumin promoted the expression of antitumor cytokines and suppressed the expression of protumor cytokines in CM-Cal27-induced Raw264.7 cells. (**A**,**B**). qRT-PCR analysis of mRNA levels of proinflammatory cytokines (TNF-α, iNOS, and IL-12) and anti-inflammatory cytokines (IL-10, Arg-1, and TGF-β) in Raw264.7 cells after stimulation with CM-Cal27. (**C**). RAW264.7 cells were first exposed to CM-Cal27 or IL-4 for the indicated time. Then, Raw264.7 cells were stimulated by curcumin for 24 h. The cell viability of the macrophages was detected using a CCK-8 assay. (**D**). After exposure to CM-Cal27, Raw264.7 cells were stimulated by curcumin at indicated concentrations. qRT-PCR analysis of mRNA levels of antitumor cytokines (TNF-α, IL-6, and CD86) and protumor cytokines (IL-10 and CD206) in Raw264.7 cells. (**E**–**G**). CM-Cal27-incubated Raw264.7 cells were stimulated with or without 20 µM curcumin. The expression of TNF-α, IL-6, and IL-10 was measured by ELISA (**E**). The percentages of CD86 (**F**) and CD206 (**G**) were measured through flow cytometry. Data are presented as the mean ± SD (*n* = 3). *p* values were determined by Student’s *t*-tests or analysis of variance (ANOVA). * *p* < 0.05; ** *p* < 0.01; *** *p* < 0.001.

**Figure 2 cells-11-03473-f002:**
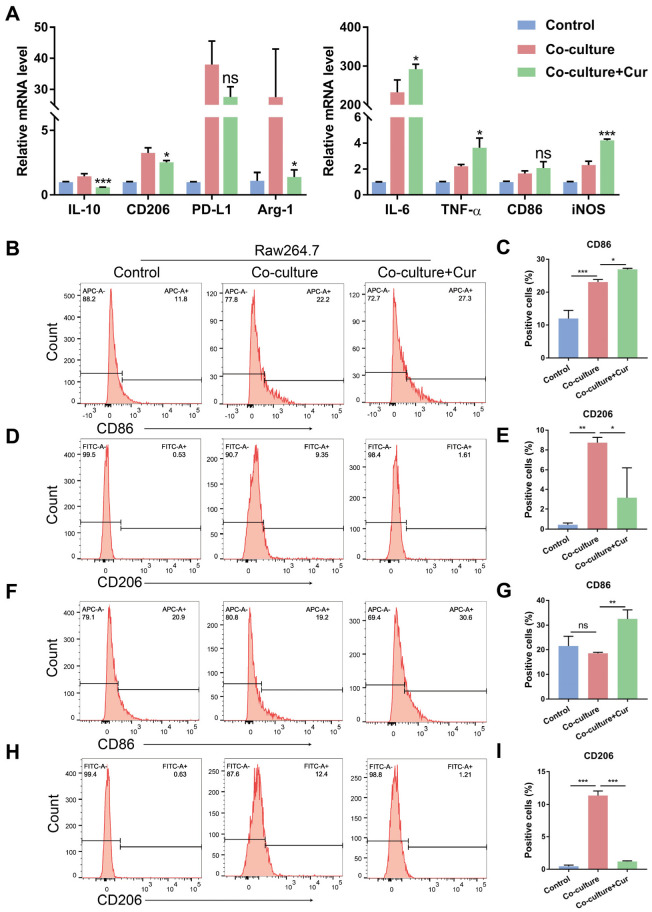
Curcumin facilitated TAMs polarization into an M1-like population in a co-culture system. Raw264.7 cells were co-cultured with Cal27 cells first, then the upper chambers were removed, and, finally, the Raw264.7 cells in the lower chambers were stimulated by curcumin (20 µM) for 24 h. (**A**). The relative mRNA expression of IL-10, Arg-1, PD-L1, CD206, CD86, TNF-α, IL-6, and iNOS was measured by qRT-PCR assay in Raw264.7 cells treated with or without curcumin. (**B**–**E**). Flow cytometric analysis of surface markers of CD86 and CD206 in Raw264.7 cells. (**F**–**I**). Raw264.7 cells were co-cultured with Cal27 cells for 24 h and then stimulated by curcumin (20 µM) without removing the Cal27 cells in the upper chambers for 24 h. Flow cytometric analysis of surface markers of CD86 and CD206 in Raw264.7 cells in the lower chambers. Data are presented as the mean ± SD (*n* = 3). *p* values were determined by one-way analysis of variance (ANOVA). * *p* < 0.05; ** *p* < 0.01; *** *p* < 0.001.

**Figure 3 cells-11-03473-f003:**
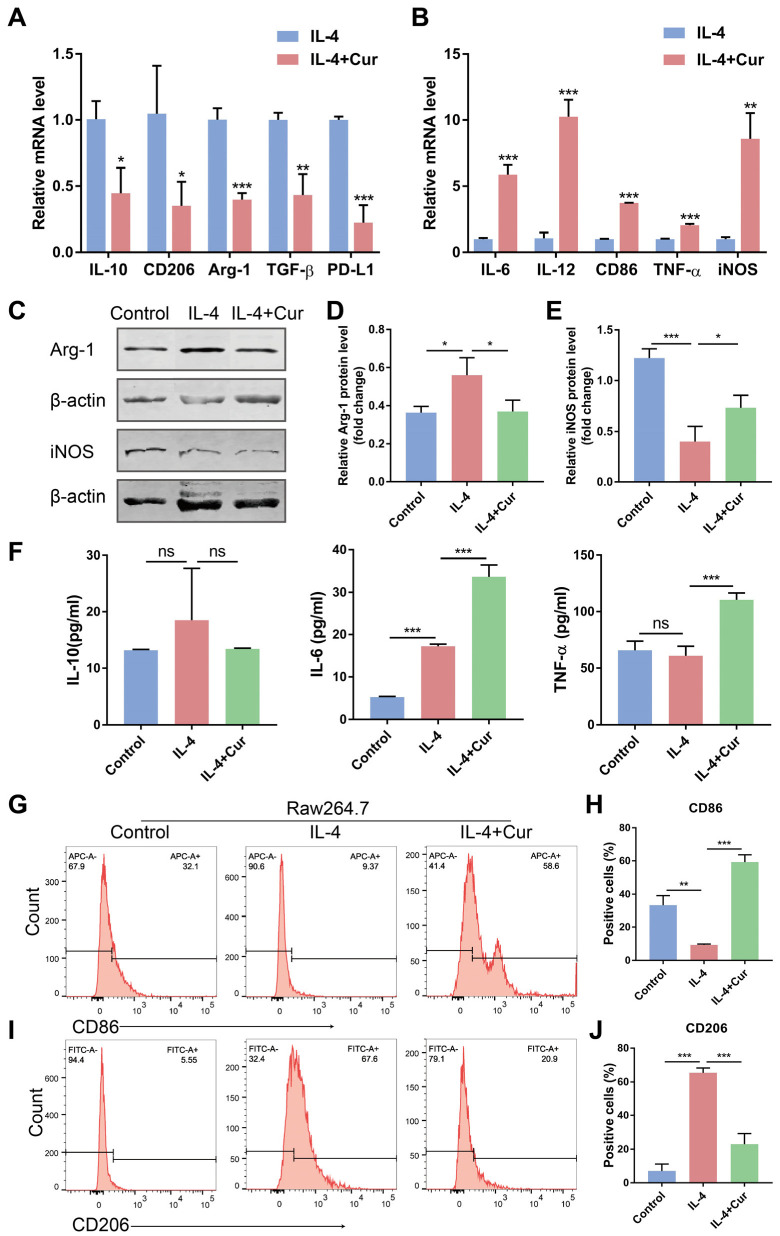
Curcumin reprogrammed M2 macrophages into an M1-like population in IL-4-induced model. Raw264.7 cells were pretreated with IL-4 (20 ng/mL) for 24 h and then stimulated by curcumin (20 µM). (**A**,**B**). The relative mRNA expressions of M1 macrophage markers TNF-α, IL-6, IL-12, CD86, and iNOS and M2 macrophage markers IL-10, CD206, TGF-β, and Arg-1 were measured by qRT-PCR assay. (**C**–**E**). The protein levels of Arg-1 and iNOS were evaluated through Western blot. (**F**). The secretion of IL-6, TNF-α, and Il-10 was measured by ELISA. (**G**–**J**). The percentage of the CD86 (**G**,**H**) and CD206 (**I**,**J**) positive populations in Raw264.7 cells. Data are presented as the mean ± SD (*n* = 3). *p* values were determined by Student’s *t*-tests or analysis of variance (ANOVA). * *p* < 0.05; ** *p* < 0.01; *** *p* < 0.001.

**Figure 4 cells-11-03473-f004:**
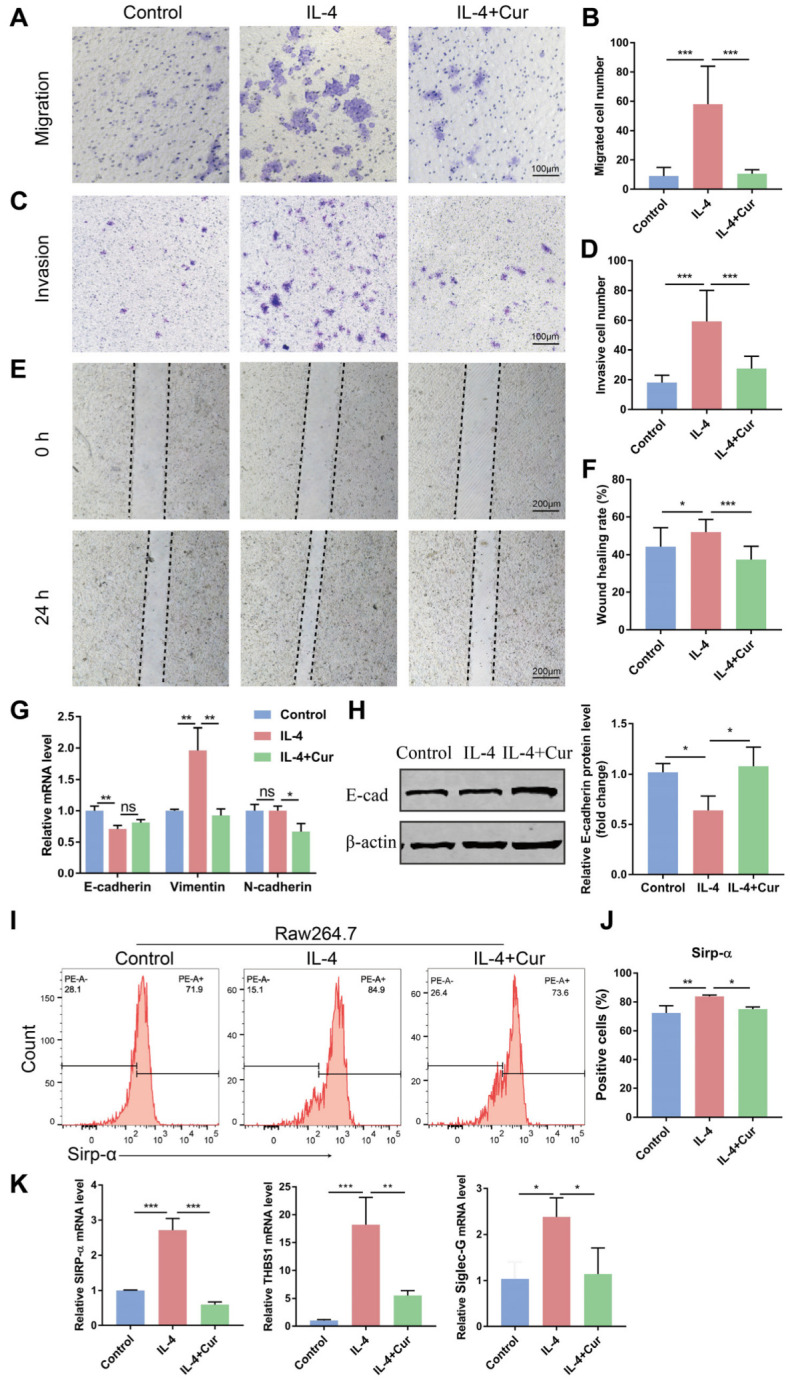
Curcumin reprogrammed M2 macrophages from a protumor phenotype towards an antitumor phenotype. IL-4 (20 ng/mL)-induced Raw264.7 cells were stimulated by curcumin (20 µM) for 24 h and further co-cultured with Cal27 cells in a 24-well Transwell plate for indicated time. Then, cells that had migrated or invaded through the membrane were fixed in 4% paraformaldehyde for 15 min and stained with 0.2% crystal violet dye for 20 min to be photographed under the microscope. (**A**,**B**). Transwell migration assays of Cal27 cells. (**C**,**D**). Transwell invasion assays of Cal27 cells. (**E**–**H**). Cal27 cells were cultured in a supernatant from M2 macrophages treated with or without curcumin. Representative micrographs of the wound-healing assay at 0 h and 24 h (**E**,**F**). The mRNA levels of Vimentin, N-cadherin, and E-cadherin in Cal27 cells were detected by qRT-PCT (**G**). The relative protein level of E-cadherin in Cal27 cells was detected through Western blot (**H**). (**I**,**J**). The surface level of Sirp-α in Raw264.7 cells was measured through flow cytometric analysis. (**K**). The mRNA levels of Sirp-α, THBS1, and Siglec-G in Raw264.7 cells were detected by qRT-PCT. Data are presented as the mean ± SD (*n* = 3). *p* values were determined by one-way analysis of variance (ANOVA). * *p* < 0.05; ** *p* < 0.01; *** *p* < 0.001.

**Figure 5 cells-11-03473-f005:**
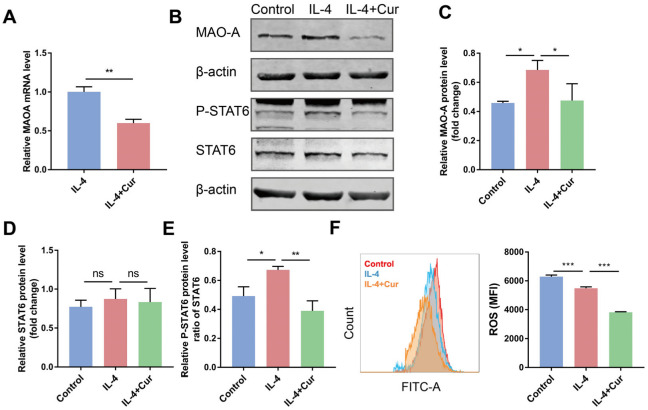
Curcumin reprogrammed M2 macrophages towards M1 macrophages by suppressing MAO-A/STAT6 signaling. Raw264.7 cells were pretreated with IL-4 (20 ng/mL) for 24 h and then stimulated by curcumin (20 µM). (**A**). The expression of MAO-A was measured by qRT-PCR. (**B**–**E**). Western blot analyses of MAO-A and STAT6 signaling in macrophages treated with or without curcumin. (**F**). Flow cytometric analysis of ROS level in Raw264.7 cells. Data are presented as the mean ± SD (*n* = 3). *p* values were determined by Student’s *t*-tests or analysis of variance (ANOVA). * *p* < 0.05; ** *p* < 0.01; *** *p* < 0.001.

**Table 1 cells-11-03473-t001:** Primers used for qRT-PCR.

Gene	Forward Primer 5′-3′	Reverse Primer 5′-3′
IL-6	ACTTCCATCCAGTTGCCTTCTTGG	TTAAGCCTCCGACTTGTGAAGTGG
IL-10	CTGCTATGCTGCCTGCTCTTACTG	ATGTGGCTCTGGCCGACTGG
IL-12	GAGGACTTGAAGATGTACCAG	TTCTATCTGTGTGAGGAGGGC
CD86	ACGGAGTCAATGAAGATTTCCT	GATTCGGCTTCTTGTGACATAC
CD206	CTCTGTTCAGCTATTGGACGC	CGGAATTTCTGGGATTCAGCTTC
PD-L1	CAGAAGCTGAGGTAATCTGGA	TGAGTCCTGTTCTGTGGAGG
iNOS	GAGACAGGGAAGTCTGAAGCAC	CCAGCAGTAGTTGCTCCTCTTC
TNF-α	CCCCAAAGGGATGAGAAGTT	CACTTGGTGGTTTGCTACGA
TGF-β	CTAATGGTGGAAACCCACAACG	TATCGCCAGGAATTGTTGCTG
Arg-1	CATTGGCTTGCGAGACGTAGAC	GCTGAAGGTCTCTTCCATCACC
E-Cadherin	CCTGGGACTCCACCTACAGAA	AGGAGTTGGGAAATGTGAGC
N-Cadherin	AACAGCAACGACGGGTTAGT	CAGACACGGTTGCAGTTGAC
Vimentin	AGGCGAGGAGAGCAGGATTT	AGTGGGTATCAACCAGAGGGA
PD-1	CGGTTTCAAGGCATGGTCATTGG	TCAGAGTGTCGTCCTTGCTTCC
Siglec-G	GAGGAGTTCAGGCTACAAGTGG	GGCATTGGTTGAAGGTCCAGGA
THBS1	GTCCACTCAGACCAGGGAGA	AAAGGTGTCCTGTCCCATCA
SIRP-α	GGCAACAAGGAGGTCACAGT	TCCGCGTCCTGTTTCTGTA
MAO-A	CCTGGTATCATGACTCTGTATGG	CTTGGACTCAGGCTCTTGAAC
GAPDH	GCACCGTCAAGGCTGAGAAC	TGGTGAAGACGCCAGT
β-actin	GTGCTATGTTGCTCTAGACTTCG	ATGCCACAGGATTCCATACC

## Data Availability

Not applicable.

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
