# Peer review of "Curcumin Reprograms TAMs from a Protumor Phenotype towards an Antitumor Phenotype via Inhibiting MAO-A/STAT6 Pathway"

_cells, 2022, doi:10.3390/cells11213473_

Round 1
Reviewer 1 Report
1- Often in the figures with the FACS analyzes There is no correspondence between the "histogram plot" and the values shown in the histogram (e.g.Fig 3D and 3E).
2- The authors demonstrate increases in protein expression of specific antigens in several figures by FACS analysis. Looking at the histogram graph, this increase is not clearly evident. probably choose the dot plot these differences are more appreciable (e.g. Fig 1 F). Moreover, for FACS analysy from ctrl to tratetd, the Gate must be the same (e.g. Fig2b-f-g; Fig3D…)
3-If the authors want to demonstrate the activity of curcumin in reprogramming macrophages from an M2 phenotype to an M1-like phenotype (expressing IL-6), how can the progressive reduction of the transcriptional level of IL-6 be explained as the concentration of curcumin increases? Maybe IL6 does not have a pivotal role?
4- If CM-Cal27 promotes polarization to M2 macrophages (expressing high levels of CD206 and low levels of CD86), why does the conditioned medium increase CD86 (FIGURE 1D) and not CD206?
5-The quality of all images (WB, histogram plots, wound healing ) are very poor
Author Response
Point 1: Often in the figures with the FACS analyzes There is no correspondence between the "histogram plot" and the values shown in the histogram (e.g.Fig 3D and 3E).
Response 1: In my original manuscript, there was a mismatch between the "histogram plot" and the values shown in the histogram (e.g.Fig 3D and 3E). This mistake had been corrected in the revised manuscript. (page 9, Figure 3G-J)
Point 2: The authors demonstrate increases in protein expression of specific antigens in several figures by FACS analysis. Looking at the histogram graph, this increase is not clearly evident. probably choose the dot plot these differences are more appreciable (e.g. Fig 1 F). Moreover, for FACS analysy from ctrl to tratetd, the Gate must be the same (e.g. Fig2b-f-g; Fig3D…).
Response 2: We have provided dot plots of FACS results in the “supplementary material of FACS” All the results of FACS have been reanalyzed, and both the control group and experimental group are the same at the gate in the revised version.
Point 3: If the authors want to demonstrate the activity of curcumin in reprogramming macrophages from an M2 phenotype to an M1-like phenotype (expressing IL-6), how can the progressive reduction of the transcriptional level of IL-6 be explained as the concentration of curcumin increases? Maybe IL6 does not have a pivotal role?
Response 3: There may be a time gap between the transcriptional level and secretion level of IL-6, this part of the experiment has been repeated. The difference is that RNA was extracted from CM-Cal27-induced Raw264.7 cells 6 h after stimulation by curcumin, rather than 24 h in the original manuscript, and the expression of IL-6 was quantified through qRT-PCR. Our results suggest that curcumin promotes the expression of M1-like phenotype marker IL-6 in CM-Cal27-incubated Raw264.7 cells at a concentration of 20 µM. (page 6, Figure 1D)
Point 4: If CM-Cal27 promotes polarization to M2 macrophages (expressing high levels of CD206 and low levels of CD86), why does the conditioned medium increase CD86 (FIGURE 1D) and not CD206?
Response 4: Treatment with the CM-Cal27 only increased the percentage of the CD86-positive population in Raw264.7 cells, but not the CD206-positive population. Curcumin affects neither the expression of the M2 macrophage surface marker CD206 nor the M1 macrophage surface marker CD86, as measured through flow cytometry (page 6, Figure 1F-G). These results suggest that Raw264.7 cells may predominantly differentiate into M1 macrophages after exposure to CM-Cal27 for 48 h. Because CM-Cal27 was filtered through a filter, this may have filtered out some extracellular vesicles with diameters over 220 nm. Furthermore, freezing and thawing affected the stabilization of CM-Cal27. These factors may affect the differentiation of Raw264.7 cells induced by CM-Cal27. Figure 1 only showed that curcumin promoted the expression of antitumor cytokines and suppressed the expression of protumor cytokines in CM-Cal27-induced Raw264.7 cells. To observe the effect of curcumin on the M1 and M2 phenotypes, we further adopted a co-culture system to stimulate Raw264.7 cells, and the results indicate that the co-culture system could motivate Raw264.7 cells to express CD86 and CD206 (page 8, Figure 2B-E). The following experiment adopted the co-culture system to further confirm that curcumin could change the phenotype of TAMs (page 8, Figure 2). Moreover, a great number of studies demonstrate that Raw264.7 cells treated with IL-4 (20 ng/ml) for 24 h can differentiate into mature M2 macrophages[1-3]. These classical polarized M2 macrophages were used to explore the effect of curcumin on the macrophage phenotype and functions in detail later in our manuscript, and our results supported that curcumin can promote polarization of IL-4-induced Raw264.7 cells towards antitumor phenotype (page 9, Figure 3; page 10, Figure 4).
1. Zhou, Q., et al., Carfilzomib modulates tumor microenvironment to potentiate immune checkpoint therapy for cancer. EMBO Mol Med, 2022. 14(1): p. e14502.
2. Yang, Y., et al., PSTPIP2 connects DNA methylation to macrophage polarization in CCL4-induced mouse model of hepatic fibrosis. Oncogene, 2018. 37(47): p. 6119-6135.
3. Dong, X., et al., Nuanxinkang protects against ischemia/reperfusion-induced heart failure through regulating IKKβ/IκBα/NF-κB-mediated macrophage polarization. Phytomedicine, 2022. 101: p. 154093.
Point 5: The quality of all images (WB, histogram plots, wound healing ) are very poor.
Response 5: We have redrawn all the Figures and re-uploaded them to the revised manuscript at high resolution.

Reviewer 2 Report
The study utilizes conditioned media from the human cell lines and add them to Mouse macrophage. the human cytokines and growth factors released by a human cell will have no or little effect on the mouse cells, forget about TAM generation. The manuscript has many flaws in the data, but it essentially doesn't qualify for publication.
Author Response
Point 1: The study utilizes conditioned media from the human cell lines and add them to Mouse macrophage. the human cytokines and growth factors released by a human cell will have no or little effect on the mouse cells, forget about TAM generation. The manuscript has many flaws in the data, but it essentially doesn't qualify for publication.
Response 1: It has been reported that human breast cancer cell line MDA-MB-231 cells and MDA-MB-231 cell-conditioned medium can induce the osteoclastic differentiation of murine Raw264.7 cells[1, 2]. MiR-200c in the MDA-MB-231 cell line increased the CD206 expression in RAW264.7 cells[3]. The interaction between CD44S expression on the human non-small cell lung cancer cell line H322 cells and OPN produced by activated Raw264.7 cells mediated the cytotoxicity of activated RAW264.7 cells against the H322 cells[4]. These results suggest that the human cytokines and growth factors may affect the mouse cells.
On the other hand, we have included new experiment results in the revised manuscript. As shown in Figure 1A-B (page 6), the conditional medium of Cal27 cells (CM-Cal27) simultaneously upregulated the relative mRNA expression of anti-inflammatory and pro-inflammatory cytokines in Raw264.7 cells at different points in time, suggesting that Raw264.7 may undergo differentiation towards TAMs. Curcumin treatment promoted the expression of antitumor cytokines (IL-6 and TNF-α) and decreased the expression of protumor cytokine (IL-10) in CM-Cal27-incubated Raw264.7 cells at a concentration of 20 µM (page 6, Figure 1D-E).
However, treatment with the CM-Cal27 only increased the percentage of the CD86-positive population in Raw264.7 cells, but not the CD206-positive population. Curcumin affects neither the expression of the M2 macrophage surface marker CD206 nor the M1 macrophage surface marker CD86, as measured through flow cytometry (page 6, Figure 1F-G). These results suggest that Raw264.7 cells may predominantly differentiate into M1 macrophages after exposure to CM-Cal27 for 48 h. Because CM-Cal27 was filtered through a filter, this may have filtered out some extracellular vesicles with diameters over 220 nm. Furthermore, freezing and thawing affected the stabilization of CM-Cal27. These factors may affect the differentiation of Raw264.7 cells induced by CM-Cal27. A great number of studies demonstrate that Raw264.7 cells treated with IL-4 (20 ng/ml) for 24 h can differentiate into mature M2 macrophages[5-7]. These classical polarized M2 macrophages were used to explore the effect of curcumin on the macrophage phenotype and functions in detail later in our manuscript, and our results supported that curcumin can promote polarization of IL-4-induced Raw264.7 cells towards antitumor phenotype.
1. Zhang, Y., et al., Sinomenine inhibits osteolysis in breast cancer by reducing IL-8/CXCR1 and c-Fos/NFATc1 signaling. Pharmacol Res, 2019. 142: p. 140-150.
2. Carina, V., et al., Inhibitory effects of low intensity pulsed ultrasound on osteoclastogenesis induced in vitro by breast cancer cells. J Exp Clin Cancer Res, 2018. 37(1): p. 197.
3. Meng, Z., et al., miR-200c/PAI-2 promotes the progression of triple negative breast cancer via M1/M2 polarization induction of macrophage. Int Immunopharmacol, 2020. 81: p. 106028.
4. Takahashi, K., et al., Restoration of CD44S in non-small cell lung cancer cells enhanced their susceptibility to the macrophage cytotoxicity. Lung Cancer, 2003. 41(2): p. 145-53.
5. Zhou, Q., et al., Carfilzomib modulates tumor microenvironment to potentiate immune checkpoint therapy for cancer. EMBO Mol Med, 2022. 14(1): p. e14502.
6. Yang, Y., et al., PSTPIP2 connects DNA methylation to macrophage polarization in CCL4-induced mouse model of hepatic fibrosis. Oncogene, 2018. 37(47): p. 6119-6135.
7. Dong, X., et al., Nuanxinkang protects against ischemia/reperfusion-induced heart failure through regulating IKKβ/IκBα/NF-κB-mediated macrophage polarization. Phytomedicine, 2022. 101: p. 154093.
Reviewer 3 Report
This is a very interesting paper that examines whether curcumin can reverse immunosuppression caused by cancer. There is a lot of interest in research that leads to immunotherapy. However, there is a lack of information for the method to replicate the experiment. Also, some results cannot be interpreted from the figures provided. Please add the following information.
l Please write “curcuma longa” and “in vitro” in italics.
l In the method, please indicate the company name of the reagent. For example, CCK-8 reagent, PrimeScriptTM RT Reagent Kit and TB Green Premix Ex Taq II kit
l When the following words “MAO-A”, “DMEM”, “FBS”, “”and C appear for the first time, please write the full name and then the abbreviation.
l Please describe the protein quantification method by Western blotting.
l Please describe the cell culture conditions for evaluating STAT phosphorylation in Western blotting experiments. Do you use phosphorylation inhibitors? which manufacturer? What is the concentration?
l Please indicate the product name and company name of the secondary antibody in Western blotting.
l Please describe the cell culture conditions and labeling conditions for flow cytometry.
l Please describe the official name of FACS.
l What are the measurement wavelengths for flow cytometry?
l When was curcumin added in the ROS experiment? When and how long was the source of active oxygen added when measuring active oxygen?
l TME appears on line 200. What is TME?
l Please add scattergram to flow site diagrams. I couldn't judge why it was classified by the indicated line.
l What was A in Fig. 4 dyed with? Add to Method and Legends.
l I couldn't read the numbers above the scale bar in Fig4B. Could you give me the resolution?
Author Response
Point 1: Please write “curcuma longa” and “in vitro” in italics.
Response 1: “Curcuma longa” and “in vitro” have been written in italics.
Point 2: In the method, please indicate the company name of the reagent. For example, CCK-8 reagent, PrimeScriptTM RT Reagent Kit and TB Green Premix Ex Taq II kit
Response 2: The company names of the reagents have been shown in the materials and methods. Such as Cell Counting Kit-8 reagent (APEXBIO, USA), PrimeScriptTM RT Reagent Kit (Takara Bio Inc., Dalian, China), and TB Green Premix Ex Taq II kit (Takara Bio Inc., Dalian, China). (page 3, line 85; page 4, lines 114-118)
Point 3: When the following words “MAO-A”, “DMEM”, “FBS”, “”and C appear for the first time, please write the full name and then the abbreviation.
Response 3: When these words appear for the first time, the full name and then the abbreviation have been included in the revised manuscript. Such as monoamine oxidase A (MAO-A), Dulbecco’s modified Eagle’s medium (DMEM), and fetal bovine serum (FBS).
Point 4: Please describe the protein quantification method by Western blotting.
Response 4: The protein quantification method by Western blotting has been added to the materials and methods. (page 4, lines 140-143)
Point 5: Please describe the cell culture conditions for evaluating STAT phosphorylation in Western blotting experiments. Do you use phosphorylation inhibitors? which manufacturer? What is the concentration?
Response 5: Raw264.7 cells were cultured with complete DMEM containing IL-4 (20 ng/ml) for 24 h to differentiate into M2 macrophages, which were further treated with curcumin (20 µM) for 24 h. Then, the cells were harvested and lysed with RIPA lysis buffer (Beyotime, P0013B) containing PMSF (Phenylmethanesulfonyl fluoride, Beyotime, ST505 ) and phosphatase inhibitor cocktail A (Beyotime, P1081) to collect total protein. The 50X store concentration of phosphatase inhibitor cocktail A includes 250 mM sodium fluoride, 50 mM sodium pyrophosphate, 50 mM b-glycerophosphate, and 50 mM sodium orthovanadate. (page 4, lines 121-128)
Point 6: Please indicate the product name and company name of the secondary antibody in Western blotting.
Response 6: The secondary antibody Dylight 800 Goat Anti-Rabbit IgG (Abbkine; A23920, 1:2,000) was used to incubate the PVDF Membrane (Millipore). (page 4, lines 138-139)
Point 7: Please describe the cell culture conditions and labeling conditions for flow cytometry.
Response 7: After stimulation with IL-4 (20 ng/ml) or co-culture with Cal27 cells, Raw264.7 cells were incubated in the medium with curcumin (20 µM) for 24 h. Then, a single-cell suspension of the Raw264.7 cells was prepared and stained with the following fluorochrome-conjugated antibodies in PBS containing 1% BSA on ice for 20 min at 4°C according to the dilution ratio recommended by the manufacturers: anti-mouse APC-CD86, anti-mouse FITC-CD20, and anti-mouse PE-Sirp-α. Fluorescence-activated cell sorting (FACS) was used to measure stained cells. (page 5, lines 150-156)
Point 8: Please describe the official name of FACS.
Response 8: The official name of fluorescence-activated cell sorting (FACS) has been included in the materials and methods where it appears for the first time. (page 5, line 155)
Point 9: What are the measurement wavelengths for flow cytometry?
Response 9: The measurement wavelengths for flow cytometry have been added. The measurement wavelength of CD206 and Sirp-α is blue laser (488 nm), and the measurement wavelength of CD86 is red laser (633 nm). (page 5, lines 156-157)
Point 10: When was curcumin added in the ROS experiment? When and how long was the source of active oxygen added when measuring active oxygen?
Response 10: After inducing differentiation using IL-4 (20 ng/ml) for 24 h, curcumin (20 µM) was added to stimulate Raw264.7 cells for another 24 h. Then, Raw264.7 cells (1´10^6 cells) were collected and resuspended in 1 ml DMEM that contained 1 µM DCFH-DA (Beyotime, S0033M). After 20 min of incubation at 37°C, the cells were immediately washed with DMEM three times and subjected to flow cytometry analysis. (page 5, lines 160-164)
Point 11: TME appears on line 200. What is TME?
Response 11: Tumor microenvironment (TME) has been included in the revised manuscript when it appears for the first time. (page 1, lines 34-35)
Point 12: Please add scattergram to flow site diagrams. I couldn't judge why it was classified by the indicated line.
Response 12: The dot plots of FACS have been provided in the “supplementary material of FACS”.
Point 13: What was A in Fig. 4 dyed with? Add to Method and Legends.
Response 13: In Figure 4A, The cells that had migrated or invaded through the membrane were fixed in 4% paraformaldehyde for 15 min and stained with 0.2% crystal violet dye for 20 min. Then, the upper chambers were washed with distilled water and photographed under the microscope. The cells that had traversed the membrane were counted in five different fields (200×). (page 3, lines 97-99; page 11, lines 271-273)
Point 14: I couldn't read the numbers above the scale bar in Fig4 B. Could you give me the resolution?
Response 14: The numbers above the scale bar have been re-added in Fig4. All the Figures have been redrawn and re-uploaded to the revised manuscript at high resolution (600ppi). (page 10)

Round 2
Reviewer 1 Report
The authors answered all doubts, even explaining the unclear results Following the suggestion the authors improved the quality off the manuscript.
Reviewer 2 Report
Accept